# Fast and covariate-adaptive method amplifies detection power in large-scale multiple hypothesis testing

Martin J. Zhang [1], Fei Xia [1] & James Zou [1,2,3]

Multiple hypothesis testing is an essential component of modern data science. In many settings, in addition to the *p*-value, additional covariates for each hypothesis are available, e.g., functional annotation of variants in genome-wide association studies. Such information is ignored by popular multiple testing approaches such as the Benjamini-Hochberg procedure (BH). Here we introduce AdaFDR, a fast and flexible method that adaptively learns the optimal *p*-value threshold from covariates to significantly improve detection power. On eQTL analysis of the GTEx data, AdaFDR discovers 32% more associations than BH at the same false discovery rate. We prove that AdaFDR controls false discovery proportion and show that it makes substantially more discoveries while controlling false discovery rate (FDR) in extensive experiments. AdaFDR is computationally efficient and allows multi-dimensional covariates with both numeric and categorical values, making it broadly useful across many applications.

[1] Department of Electrical Engineering, Stanford University, Palo Alto 94304, USA. [2] Department of Biomedical Data Science, Stanford University, Palo Alto 94304, USA. [3] Chan-Zuckerberg Biohub, San Francisco 94158, USA. Correspondence and requests for materials should be addressed to J.Z. (email: jamesz@stanford.edu)

Multiple hypothesis testing (or multiple testing correction) is an essential component in many modern data analysis workflows. A common objective is to maximize the number of discoveries while controlling the fraction of false discoveries (FD). For example, we may want to identify as many genes as possible that are differentially expressed between two populations such that less than, say, 10% of these identified genes are false positives.

In the standard setting, the data for each hypothesis is summarized by a p-value, with a smaller value presenting stronger evidence against the null hypothesis that there is no association. Commonly-used procedures such as the Benjamini-Hochberg procedure (BH)[1] works solely with this list of p-values[2–6]. Despite being widely used, these multiple testing procedures fail to utilize additional information that is often available in modern applications but not directly captured by the p-value.

For example, in expression quantitative trait loci (eQTL) mapping or genome-wide association studies (GWAS), single nucleotide polymorphisms (SNPs) in active chromatin states are more likely to be significantly associated with the phenotype[7]. Such chromatin information is readily available in public databases[8], but is not used by standard multiple hypothesis testing procedures—it is sometimes used for post hoc biological interpretation. Similarly, the location of the SNP, its conservation score, etc., can alter the likelihood for the SNP to be an eQTL. Together such additional information, called covariates, forms a feature representation of the hypothesis; this feature vector is ignored by the standard multiple hypothesis testing procedures.

Here we present AdaFDR, a fast and flexible method that adaptively learns the decision threshold from covariates to significantly improve the detection power while having the false discovery proportion (FDP) controlled at a user-specified level. A schematic diagram for AdaFDR is shown in Fig. 1. AdaFDR takes as input a list of hypotheses, each with a p-value and a covariate vector; it outputs a set of selected (also called rejected) hypotheses. Conventional methods like BH and Storey-BH (SBH)[3] use only p-values and have the same p-value threshold for all hypotheses (Fig. 1 top right). However, as illustrated in the bottom-left panel, the data may have an enrichment of small p-values for certain values of the covariate, which suggests an enrichment of alternative hypotheses around these covariate values. Intuitively, allocating more false discovery rate (FDR) budget to hypotheses with such covariate values could increase the detection power. AdaFDR adaptively learns such a pattern using both p-values and covariates, resulting in a covariate-dependent threshold that makes more discoveries under the same FDP constraint (Fig. 1 bottom right).

AdaFDR learns the covariate-dependent threshold by first fitting a mixture model using an expectation-maximization (EM) algorithm, where the mixture model is a combination of a generalized linear model (GLM) and Gaussian mixtures[9–11]. Then it makes local adjustments in the p-value threshold by optimizing for more discoveries. The standard assumption of AdaFDR and other related methods is that the covariates should not affect the p-values under the null hypothesis; we prove that AdaFDR controls FDP under such assumption (see the "Methods" section). AdaFDR is developed to be fast and flexible—it can simultaneously process more than 100 million hypotheses within an hour and allows multi-dimensional covariates with both numeric and categorical values. In addition, AdaFDR provides exploratory plots visualizing how each covariate is related to the significance of hypotheses, allowing users to interpret their findings. We also provide a much faster but slightly less powerful version, AdaFDR-fast, which uses only the EM step and skips the subsequent optimization. It can process more than 100 million hypotheses in around 5 min on a standard laptop.

AdaFDR is the mature development of and subsumes a previous, preliminary method that we called NeuralFDR[12]. Instead of using a neural network to model the discovery threshold as in NeuralFDR, AdaFDR uses a mixture model that lacks some flexibility but is much faster to optimize. Among other related methods[13–20], IHW[16,17] groups the hypotheses into a pre-specified number of bins and applies a constant threshold for each bin to maximize the discoveries. It is practical, well-received by the community, and can scale up to one billion hypotheses. Yet it only supports the covariate to be univariate and uses a stepwise-constant function for the threshold, which limits its detection power. Boca and Leek[20] proposes a regression framework (referred to as BL) to estimate the null proportion conditional on the covariate, and perform multiple testing via weighting the BH-adjusted p-values by their corresponding estimated null proportion. It is fast and flexible, but the method does not utilize the covariate-dependent alternative distribution information which could reduce power; it does not provide theoretical results on FDR control either. AdaPT[15] cleverly uses a p-value masking procedure to control FDR. While IHW needs to split the hypotheses into multiple folds for FDR control, AdaPT can learn the threshold using virtually the entire data and therefore has a higher power. However, such p-value masking procedure takes many iterations of optimization and can be computationally expensive. AdaFDR is designed to achieve the best of both worlds: it has a speed comparable to IHW and BL while using a flexible modeling strategy to have greater detection power than AdaPT. Some other related works include non-adaptive p-value weighting[21–24], estimation of the covariate-dependent null proportion[25,26], and estimation of the local FDR[27–31]. There are also application-specific methods that utilize the domain knowledge to increase the detection power, such as GSEA[32] for gene pathway analysis, TORUS[33] for eQTL study, and StructFDR[34] for microbiome-wide multiple testing.

We systematically evaluate the performance of AdaFDR across multiple datasets. We first consider the problem of eQTL mapping using the data from the genotype-tissue expression (GTEx) project[7,35]. As covariates, we consider the distance between the SNP and the gene, the gene expression level, the AAF as well as the chromatin states of the SNP. Across all 17 tissues considered in the study, AdaFDR has an improvement of 32% over BH and 27% over the state-of-art IHW[16,17]. We next consider other applications, including differential expression analysis for three RNA-Seq datasets[36–38] with the gene expression level as the covariate, differential abundance analysis for two microbiome datasets[39,40] with ubiquity (proportion of samples where the feature is detected) and the mean nonzero abundance as covariates, differential abundance analysis for a proteomics dataset[16,41] with the peptides level as the covariate, and signal detection for two fMRI datasets[42,43] with the Brodmann area label[44] as the covariate that represents different functional regions in human brain. In all experiments, AdaFDR shows a similar improvement. Finally, we perform extensive simulations to demonstrate that AdaFDR has the highest detection power while controlling FDP in various cases where the p-values may be either independent or dependent.

## Results

**Discovering eQTLs in GTEx**. We first consider detecting eQTLs using data from GTEx[7,35]. The GTEx project has collected both the genetic variation data (SNPs) and the gene expression data (RNA-Seq) from 44 human tissues, with sample sizes ranging from 70 (uterus) to 361 (muscle skeletal). Its goal is to study the associations between genotype and gene expression across humans. Each hypothesis test is to test if there is a significant

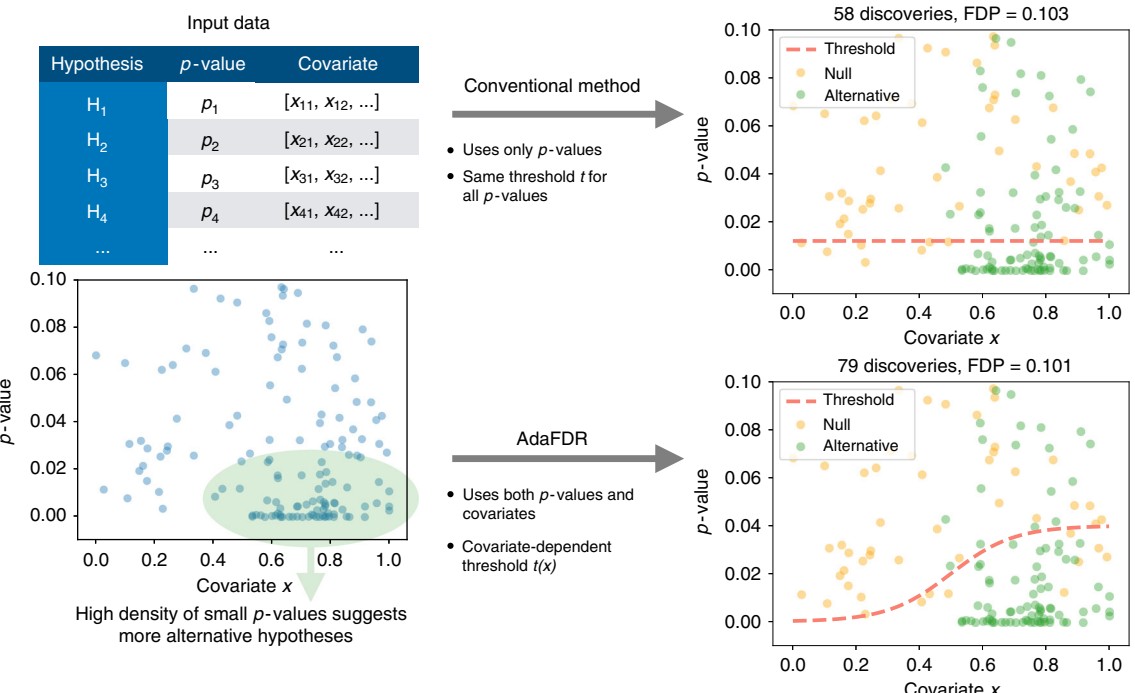

**Fig. 1** Intuition of `AdaFDR`. Top-left: As input, `AdaFDR` takes a list of hypotheses, each with a *p*-value and a covariate that could be multi-dimensional. Bottom-left: A toy example with a univariate covariate. The enrichment of small *p*-values in the bottom-right corner suggests that there are more alternative hypotheses there. Leveraging this structure can lead to more discoveries. Top-right: Conventional methods use only *p*-values and have the same *p*-value threshold for all hypotheses. Bottom-right: `AdaFDR` adaptively learns the uneven distribution of the alternative hypotheses, and makes more discoveries while controlling the false discovery proportion (FDP) at the desired level (0.1 in this case)

association between a SNP and a gene, also referred to as an eQTL. A standard caveat is that a selected eQTL (either through small *p*-value or an FDR procedure) may not be a true *causal* SNP —it could tag a nearby causal SNP due to linkage disequilibrium. We should interpret the selected eQTLs with care; nonetheless, it is still valuable to discover candidate associations and local regions with strong associations while controlling FDR[16].

We focus on cis-eQTLs where the SNP and the gene are close to each other on the genome (<1 million base pairs). Previous works provide evidence that various covariates could be associated with the significance of cis-eQTLs[7,33,35,45–47]. In this study, we consider four covariates for each SNP-gene pair: (1) the distance from SNP to gene transcription start site (TSS); (2) the log10 gene expression level; (3) the alternative allele frequency (AAF) of the SNP; 4) the chromatin state of the SNP. Out of 44 tissues, we selected 17 whose chromatin state information is available[8] and have more than 100 biological samples. For each tissue, *p*-values for all associations are tested simultaneously with numbers of hypotheses ranging from 140 to 180 million for different tissues, imposing a very-large-scale multiple hypothesis testing problem. We use a nominal FDR level of 0.01. Such an experiment of testing all SNP-gene pairs simultaneously is a prescreening step for detecting casual eQTLs and is also performed in some recent works[16,17]. A similar analysis workflow is to first discover significant genes (eGenes) and then match significant SNPs (eVariants) for each eGene[33]. There are also works that, given the eQTL discoveries, prioritize the casual SNPs based on regulatory annotations in a post-hoc fashion[45] or use eQTL findings to help identify casual SNPs in GWAS[47].

As shown in Fig. 2a, `AdaFDR` and its fast version (`AdaFDR-fast`) consistently make more discoveries than other methods in every tissue. On average, it has an improvement of 32% over BH and 27% over IHW (see Supplementary Fig. 3 for testing using

each covariate separately). Next, we investigate whether using the eQTL *p*-values of an existing tissue could boost the power of discovering eQTLs in a new tissue. To simulate this scenario, we consider specifically the two adipose tissues, Adipose_Subcutaneous and Adipose_Visceral_Omentum. For each of them, we use the −log10 *p*-values from the other tissue as an additional covariate—e.g., for Adipose_Subcutaneous, the −log10 *p*-value of Adipose_Visceral_Omentum is used as an extra covariate. Leveraging previous eQTL results substantially increases discovery power (Fig. 2b); the *p*-value augmentation (`AdaFDR` (aug)) yields 56% and 83% more discoveries for the two adipose tissues compared to BH. We then perform a control experiment, where the augmented *p*-values, instead of coming from the other adipose tissue that is similar to the one under investigation, are from a brain tissue (Brain_Caudate_basal_ganglia) that is very different from the adipose tissue (Fig. 2 in the GTEx paper[7]). In this case, the improvement in the number of discoveries due to the extra covariate vanishes for the two tissues (`AdaFDR` (ctrl)), which is consistent with the idea that `AdaFDR` learns to leverage shared genetic architecture in closely related tissues to improve power. This analysis suggests that we can potentially greatly improve eQTL discovery by leveraging related tissues during multiple hypothesis testing. We provide additional supporting experiments for the two colon tissues in Supplementary Fig. 1a.

`AdaFDR` also characterizes how each covariate affects the significance level of the hypotheses. The results for Adipose_-Subcutaneous are shown in Fig. 2c as an example. We first consider the distance from TSS and the top-left panel provides a simple visualization, where for each hypothesis (downsampled to 10 k), the *p*-values are plotted against the distances from TSS. There is a strong enrichment of small *p*-values when the distance is close to 0, indicating that the SNP and gene are more likely to have a significant association if they are close to each other. In the

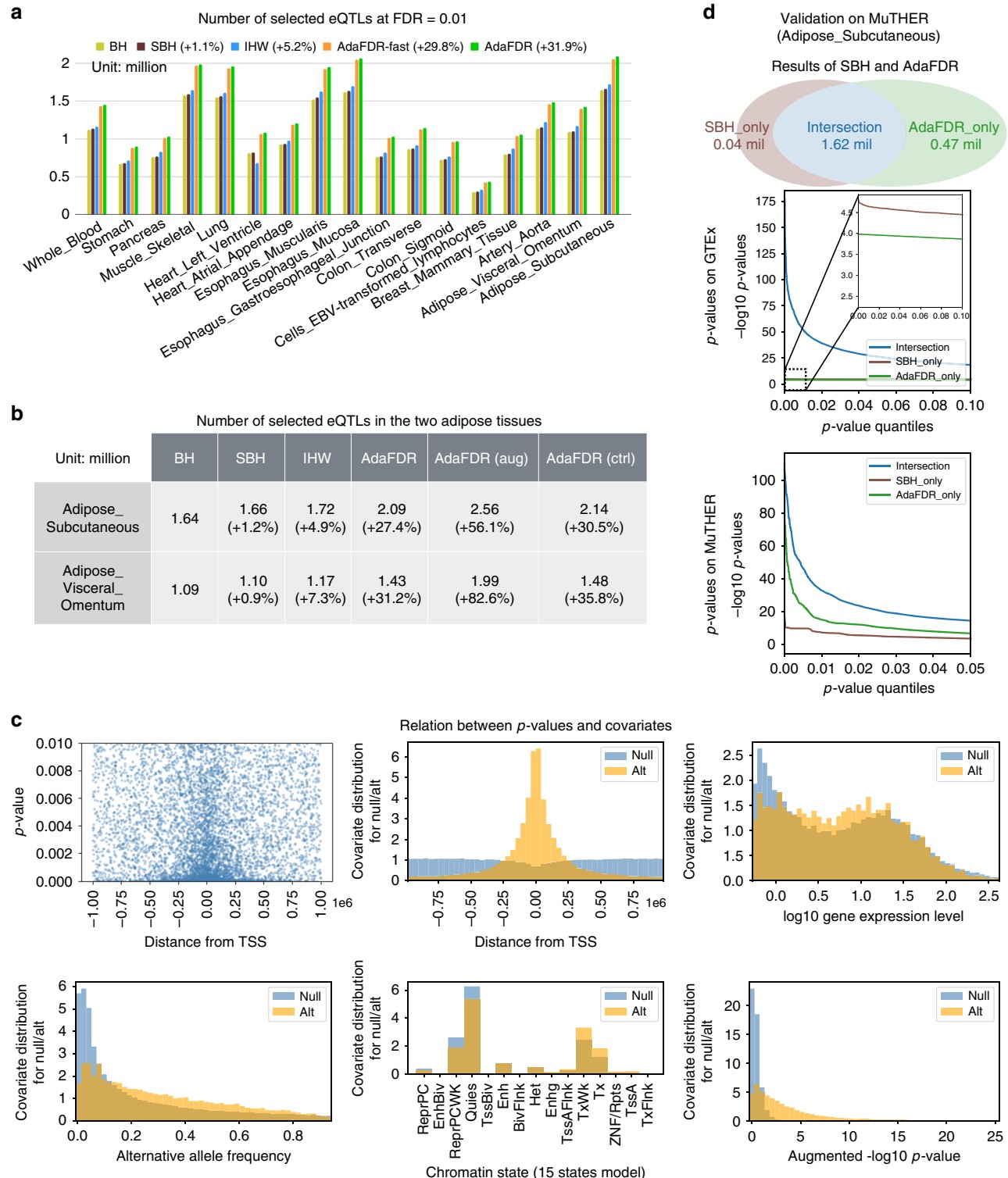

**Fig. 2** Analysis of the GTEx data. **a** Results of 17 tissues considered in the study. `AdaFDR` and its fast version `AdaFDR-fast` consistently make more discoveries than other methods. Source data are provided as a Source Data file. **b** Results on the two adipose tissues where the −log10 $p$-value from another tissue was added as an extra covariate. Using $p$-values from a similar tissue (`AdaFDR (aug)`) yields significantly more discoveries than using $p$-values from an unrelated tissue (`AdaFDR (ctrl)`). **c** Top-left: $p$-values (y-axis) plotted against the distances from TSS (x-axis); each dot corresponds to one SNP-gene pair. Small $p$-values at the center suggest that there is an enrichment of significant associations when the distance from TSS is small. Other panels: `AdaFDR`-estimated marginal covariate distribution for the null hypotheses (blue) and the alternative hypotheses (orange). Higher values of the orange distribution suggest an enrichment of alternative hypotheses. **d** Top: Discoveries made by SBH and `AdaFDR`. Middle: The $p$-values of these discoveries—SBH-only $p$-values are smaller than `AdaFDR`-only $p$-values on GTEx. Bottom: The $p$-values of the same set of discoveries on the independent MuTHER data, where `AdaFDR`-only $p$-values are smaller than SBH-only $p$-values, suggesting that `AdaFDR`-only discoveries are more likely to be true discoveries

top-center panel, `AdaFDR` characterizes such relationship by providing estimates of the marginal covariate distribution for the null hypotheses (blue) and the alternative hypotheses (orange) respectively. It learns that the distance is smaller for significant associations, consistent with previous works[7,35].

`AdaFDR` interprets other covariates in a similar fashion. Figure 2c top-right panel indicates that genes expression levels are higher for significant associations, in agreement with previous observations[15,16]. SNPs also have AAF closer to 0.5 for significant associations. In addition, the bottom-center panel indicates that for significant associations, there are more SNPs in active chromatin states—Tx (strong transcription), TxWk (weak transcription), TssA (active TSS)—as compared to inactive states—Quies (quiescent), ReprPC (repressed PolyComb) ReprPCWk (weak repressed PolyComb). Finally, the bottom-right panel shows that p-values from the augmented tissue Adipose_Visceral_Omentum are positively correlated with the significance of the associations. See Supplementary Fig. 1b for similar results on the Colon_Sigmoid tissue.

We use adipose eQTL data from the Multiple Tissue Human Expression Resource (MuTHER) project[48] to validate our GTEx eQTL discoveries. The participants in MuTHER are disjoint from the GTEx participants, making MuTHER an independent dataset. For this analysis, we compare the testing results of `AdaFDR` on Adipose_Subcutaneous with that of Storey-BH (SBH), which is known to be a better baseline than BH. As shown in the top panel of Fig. 2d, `AdaFDR` detects almost all discoveries made by SBH while having 26% more discoveries (see results for all 17 tissues in Supplementary Fig. 2a and covariate distribution of these discoveries in Supplementary Fig. 2b). The p-values of these discoveries are shown in the middle panel of Fig. 2d, where the x-axis is the p-value quantile and the y-axis is the −log10 p-value. Hypotheses discovered by both methods have significantly smaller GTEx p-values while SBH-only p-values are smaller than `AdaFDR`-only p-values in the GTEx data; the latter is due to the fact that SBH uses the same threshold for all p-values. On the MuTHER validation data, the eQTLs discovered only by `AdaFDR` have more significant p-values than the eQTLs discovered only by SBH. This reveals a counter-intuitive behavior of `AdaFDR`: it rejects some hypotheses with larger p-values if these SNPs have covariates that indicate a higher likelihood of eQTL. The MuTHER data validates this strategy—`AdaFDR` is able to discover more eQTLs on GTEx and the discovered eQTLs have more significant replication results on MuTHER. See more results on using the MuTHER adipose eQTL data to validate GTEx Adipose_Visceral_Omentum discoveries and using the MuTHER lymphocytes (LCL) eQTL data to validate GTEx Cells_EBV-transformed_lymphocytes discoveries in Supplementary Fig. 1c.

`AdaFDR` can be broadly applied to any multiple testing problem where we have covariates for the hypotheses. This includes many high-throughput biological studies beyond eQTL. Here we evaluate its applications to RNA-Seq, microbiome, proteomics and fMRI imaging data. In all cases, `AdaFDR` significantly outperforms current state-of-the-art methods.

**Small GTEx data**. AdaPT and BL cannot be run on the full GTEx data due to their computational limitations. In order to perform a direct comparison, we created a small GTEx data that contains the first 300 k associations from chromosome 21 for the two adipose tissues. This small data takes AdaPT around 15 h to process compared to less than 20 min for `AdaFDR`. As shown in Fig. 3a, `AdaFDR` has the highest number of discoveries in both experiments while AdaPT has slightly fewer. In addition, all covariate-adaptive methods except BL have significant improvement over the non-adaptive methods (BH, SBH).

**RNA-Seq data**. We considered three RNA-Seq datasets that were used for differential expression analysis in AdaPT and IHW, i.e., the Bottomly data[37], the Pasilla data[38] and the airway data[36]. Here, the log expression level is used as the covariate, and the FDR level is set to be 0.1. The results are shown in Fig. 3a, where `AdaFDR` and AdaPT have a similar number of discoveries (`AdaFDR` is consistently higher), and both are substantially more powerful than others. All covariate-adaptive methods make significantly more discoveries than the non-adaptive methods. In addition, the covariate patterns learned by `AdaFDR` are shown in Fig. 3b for the Bottomly data and the Pasilla data, and in Supplementary Fig. 6c for the airway data. The gene expression levels are higher for significantly differentially expressed genes, consistent with previous findings[15–17].

**Microbiome data**. We considered a subset of microbiome data from the Ecosystems and Networks Integrated with Genes and Molecular Assemblies (ENIGMA), where samples were acquired from monitoring wells in a site contaminated by former waste disposal ponds and all sampled wells have various geochemical and physical measurements[39,40]. Following the original study, we performed two experiments to test for correlations between the operational taxonomic units (OTUs) and the pH, Al respectively. Ubiquity and the mean nonzero abundance are used are covariates, where the ubiquity is defined as the proportion of samples in which the OTU is present. The FDR level is set to be 0.2 for more discoveries and the fast version of `AdaFDR` is used due to the small sample size. As shown in Fig. 3a, `AdaFDR` is significantly more powerful than other methods. The covariates are visualized in Fig. 3c for the pH test and Supplementary Fig. 6b for the Al test. Both the ubiquity and the mean nonzero abundance are higher for significant microbiomes. This may be because a higher level of these two quantities improves the detection power similar to the expression level in the RNA-Seq case.

**Proteomics data**. We considered a proteomics dataset where yeast cells treated with rapamycin were compared to yeast cells treated with dimethyl sulfoxide ($2 \times 6$ biological replicates)[16,41]. Differential abundance of 2,666 proteins is evaluated using Welch's t-test. The total number of peptides is used as covariate that is quantified across all samples for each protein. The FDR level is set to be 0.1 and the fast version of `AdaFDR` is used due to the small sample size. As shown in Fig. 3a, `AdaFDR` and BL have similar performance and are significantly more powerful than other methods. The covariate is visualized in Fig. 3d where the peptides levels are higher for significant proteins. This is expected since the peptides level is similar to the expression level in the RNA-Seq data.

**fMRI data**. We considered two functional magnetic resonance imaging (fMRI) experiments where the human brain is divided spatially into isotropic voxels and the null hypothesis for each voxel is that there is no response to the stimulus[42]. The first experiment was done on a single participant with auditory stimulus and the second was done on a healthy adult female participant where the stimulus was to ask the person to imagine playing tennis[43]. We use the Brodmann area label, which represents different functional regions of the human brain[44], as covariate for each voxel. The FDR level is set to be 0.1 and the fast version of `AdaFDR` is used because there is an inflation of p-values at 1 and, as a result, the mirror estimator would not function properly for the optimization step. As shown in Fig. 3a, `AdaFDR` is significantly more powerful than other methods. The result of AdaPT is omitted since it does not support categorical covariates, and directly running the GAM model yields a result

**a**

| | BH | SBH | AdaPT | IHW | BL | AdaFDR |
|---|---|---|---|---|---|---|
| **Results in other applications** | | | | | | |
| Small_GTEx: Adipose_ Subcutaneous | 1182 | 1188 (+1%) | 1333 (+13%) | 1333 (+13%) | 1185 (+0%) | **1469 (+24%)** |
| Small_GTEx: Adipose_ Visceral_Omentum | 549 | 553 (+1%) | 1037 (+89%) | 724 (+32%) | 558 (+2%) | **1360 (+148%)** |
| RNA-Seq: Bottomly | 1583 | 1693 (+7%) | 2109 (+33%) | 1714 (+8%) | **2347 (+48%)** | 2144 (+35%) |
| RNA-Seq: Pasilla | 687 | 687 (+0%) | 853 (+24%) | 785 (+14%) | 740 (+8%) | **856 (+25%)** |
| RNA-Seq: airway | 4079 | 4079 (+0%) | 6045 (+48%) | 4862 (+19%) | 4792 (+18%) | **6050 (+48%)** |
| Microbiome: enigma_ph | 61 | 65 (+7%) | 96 (+57%) | 90 (+43%) | 104 (+71%) | **124 (+103%)** |
| Microbiome: enigma_al | 206 | 437 (+112%) | 496 (+141%) | 283 (+37%) | 460 (+123%) | **503 (+144%)** |
| Proteomics | 244 | 358 (+47%) | 384 (+57%) | 245 (+0%) | 406 (+66%) | **409 (+68%)** |
| fMRI: auditory | 888 | 888 (+0%) | – | 1015 (+14%) | 889 (+0%) | **1045 (+18%)** |
| fMRI: imagination | 2141 | 2228 (+4%) | – | 2151 (+1%) | 2143 (+0%) | **2237 (+5%)** |

**b**

**c**

**d**

**e**

**Fig. 3** Results on other applications. **a** The number of discoveries of various methods on two small GTEx eQTL datasets, three RNA-Seq datasets, two microbiome datasets, one proteomics dataset, and two fMRI datasets. `AdaFDR` is used for the small GTEx and the RNA-Seq datasets while `AdaFDR-fast` is used for others, due to their smaller data size. The fMRI results for AdaPT are omitted since the AdaPT software does not support categorical covariates. **b** Covariate visualization for RNA-Seq datasets. **c** Covariate visualization for the microbiome dataset. **d** Covariate visualization for the proteomics dataset. **e** Covariate visualization for fMRI datasets

much worse than BH. The covariate is visualized in Fig. 3e. For the auditory experiment, the Brodmann areas corresponding to auditory cortices, namely 41, 42, 22, are among areas enriched with significant discoveries. For the tennis imagination experiment, multiple cortices seem to respond to this stimulus, including auditory cortex (42), visual cortices (18, 19), and motor cortices (4, 6, 7).

**Simulation studies**. In order to systematically quantify the FDP and power of all methods, we conducted extensive analyses of synthetic data where we know the ground truth. Each experiment is repeated 10 times and 95% confidence intervals are provided. In Fig. 4a, the top two panels correspond to a simulated data with one covariate while the bottom two panels correspond to a

simulated data with weakly-dependent *p*-values generated according to a previous paper[4]. In both simulations, all methods control FDR while `AdaFDR` has significantly higher power. Additional simulation experiments with strongly-dependent *p*-values and higher dimensional covariates can be found in Supplementary Fig. 8a, where similar results are observed. Detailed descriptions of the synthetic data can be found in Supplementary Note 2.

We also investigate the running time of different methods. In Fig. 4b, all experiments are repeated 5 times and the 95% confidence intervals are provided. The top panel uses a simulated dataset with a 2d covariate, with the number of hypotheses varying from 20 to 100 k. `AdaFDR-fast` takes 10 s to run while `AdaFDR`, IHW and BL finished within a reasonable time of

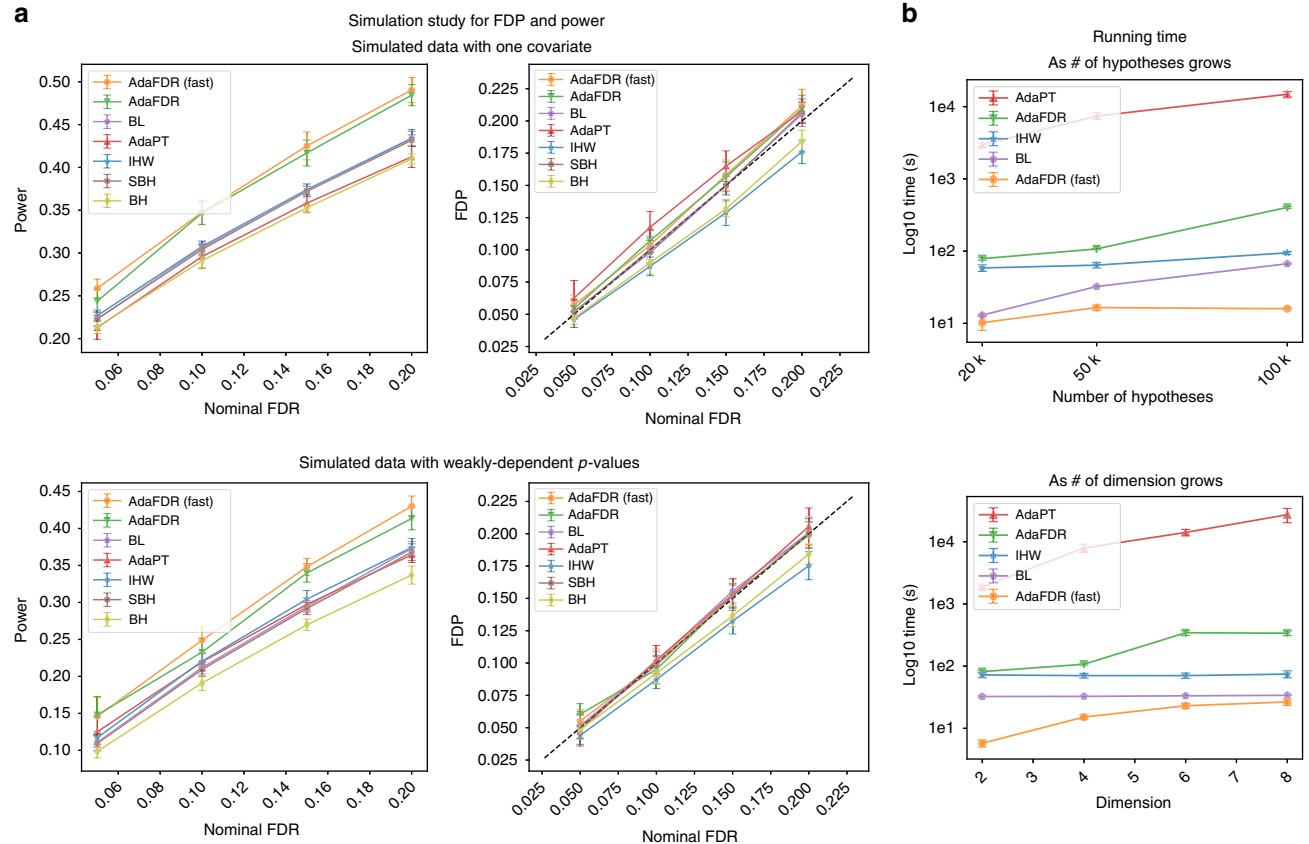

**Fig. 4** Simulation studies. 95% confidence intervals are provided for all panels. **a** Simulation of FDP and power on an independent case (top) and a weakly-dependent case (bottom). **b** Running time analysis. Top: as the number of hypotheses grows. Bottom: as the covariate dimension grows

around 100 s. AdaPT, however, needs a few hours to finish, significantly slower than other methods. In the bottom panel, the number of hypotheses is fixed to be 50 k and the covariate dimension varies from 2 to 8; a similar result is observed.

We also consider the main simulation benchmark in a recent comparison paper[40] that includes two RNA-Seq in silico experiments, one experiment with an uninformative covariate, and another two experiments that vary the number of hypotheses and the null proportion respectively. We run AdaFDR on this benchmark without any modification or tuning; AdaFDR achieves greater power than all other methods while controlling FDR (Supplementary Figs. 9 and 10). AdaFDR reduces to SBH when the covariate is not informative, indicating that it is not overfitting the uninformative covariate (Supplementary Figs. 9e and 10e).

**Comparison with NeuralFDR.** AdaFDR is the mature development of and subsumes a previous, preliminary method that we called NeuralFDR[12]. Instead of using a neural network to model the discovery threshold as in NeuralFDR, AdaFDR uses a mixture model that lacks some flexibility but is much faster to optimize—for the GTEx data used in the NeuralFDR paper, it takes NeuralFDR 10+ hours to process but only 9 min for AdaFDR. Yet, AdaFDR maintains a similar discovery power on the benchmark data used to test NeuralFDR (Supplementary Fig. 8b).

**Discussion**
Here we propose AdaFDR, a fast and flexible method that efficiently utilizes covariate information to increase detection power.

Extensive experiments show that AdaFDR has greater power than existing approaches while controlling FDR. We discuss some of its characteristics and limitations.

Our theory proves that AdaFDR controls FDP in the setting when the null hypotheses are independent (the alternative hypotheses can have arbitrary correlations, see Theorem 1). This is a standard assumption also used in BH, SBH, IHW and AdaPT. In practice, the user can make p-value histograms stratified by covariates as diagnostic plots to check the model assumption (Supplementary Figs. 4 and 5). To investigate the robustness of AdaFDR when there is a model mismatch, we have performed systematic simulations with different p-value correlation structures to demonstrate that AdaFDR still controls FDP even when the null hypotheses are not independent. Moreover, although there are correlations among SNPs in the eQTL study, we show that the discoveries made by AdaFDR on the GTEx data replicate well on the independent MuTHER data with a different cohort. These suggest that AdaFDR behaves well when there is a dependency between null p-values. Since none of the other methods popular methods—BH, SBH, IHW, AdaPT—provides FDR control under arbitrary dependency, our comparison experiments are fair. AdaFDR can potentially be extended to allow for controlling FDR under arbitrary dependency using a similar idea as discussed in the extended IHW paper[17] (see Supplementary Note 1.4 for more details).

The typical use-case for AdaFDR is when there are many hypotheses to be tested simultaneously—ideally more than 10 k. This is because AdaFDR needs many data to learn the covariate-adaptive threshold and to have an accurate estimate of FDP. A similar recommendation on the number of hypotheses is also made for IHW. When there are fewer hypotheses (<10 k) or it is

expected to have very few discoveries (less than a few hundreds), `AdaFDR-fast` is recommended as a more robust choice. `AdaFDR-fast` is also recommended when there is an inflation of *p*-values at 1 since the mirror estimator for the optimization step would produce an overly conservative result in this case. In addition, `AdaFDR` may produce slightly different results (<10%) in two runs with different random seeds because of the random hypotheses splitting step (see Supplementary Fig. 7 for more details). It is recommended to fix the random seed for better reproducibility. Nonetheless, in all cases the discoveries are valid in that the FDR is controlled.

The scalability of `AdaFDR` and its ability to handle multivariate discrete and continuous covariates makes it broadly applicable to any multiple testing applications where additional information is available. While we focus on genomics experiments in this paper —because most of the previous methods were also evaluated on genomics experiments—it would be interesting to also apply `AdaFDR` to other domains such as imaging association analysis.

## Methods

**Definitions and notations.** Suppose we have $N$ hypothesis tests and each of them can be characterized by a *p*-value $P_i$, a $d$-dimensional covariate $\mathbf{x}_i$, and an indicator variable $h_i$ with $h_i = 1$ representing the hypothesis to be true alternative. Then the set of true null hypotheses $\mathcal{H}_0$ and the set of true alternative hypotheses $\mathcal{H}_1$ can be written as $\mathcal{H}_0 \overset{\text{def}}{=} \{i : i \in [N], h_i = 0\}$ and $\mathcal{H}_1 \overset{\text{def}}{=} \{i : i \in [N], h_i = 1\}$, where we adopt the notation $[N] \overset{\text{def}}{=} \{1, 2, \cdots, N\}$. Given a threshold function $t(\mathbf{x})$, we reject the $i$th null hypothesis if $P_i \le t(\mathbf{x}_i)$. The number of discoveries $\mathrm{D}(t)$ and the number of false discoveries $\mathrm{FD}(t)$ can be written as $\mathrm{D}(t) \overset{\text{def}}{=} \sum_{i \in [N]} \mathbb{I}_{\{P_i \le t(\mathbf{x}_i)\}}$ and $\mathrm{FD}(t) \overset{\text{def}}{=} \sum_{i \in \mathcal{H}_0} \mathbb{I}_{\{P_i \le t(\mathbf{x}_i)\}}$. The FDP is defined as $\mathrm{FDP}(t) \overset{\text{def}}{=} \frac{\mathrm{FD}(t)}{\mathrm{D}(t) \vee 1}$, where $a \vee b \overset{\text{def}}{=} \max(a, b)$. The expected value of FDP is FDR[1]: $\mathrm{FDR} = \mathbb{E}[\mathrm{FDP}]$.

**Multiple testing via `AdaFDR`.** `AdaFDR` can take as input a multi-dimensional covariate $\mathbf{x}$. The key assumption is that the null *p*-values remain uniform regardless of the covariate value while others, including the alternative *p*-values and the likelihood for the hypotheses to be true null/alternative, may have arbitrary dependencies on the covariate. This is a standard assumption in the literature[1,15,16]. For example, in the case of AAF, the null *p*-values are uniformly distributed independent of AAF since the gene expression has no association with the SNP under the null hypothesis. However, the alternative *p*-values may depend on AAF since the associations are easier to detect/yield smaller *p*-values if the AAF is close to 0.5.

`AdaFDR` aims to optimize over a set of decision rules $t(\mathbf{x}) \in \mathcal{T}$ to maximize the number of discoveries, subject to the constraint that the FDP is less than a user-specified nominal level $\alpha$. Conceptually, this optimization problem can be written as

$$\text{maximize}_{t \in \mathcal{T}} \mathrm{D}(t),\ s.t.\ \mathrm{FDP}(t) \le \alpha. \tag{1}$$

There are three challenges in this optimization problem: (1) the set of decision thresholds $\mathcal{T}$ needs to be parameterized in such a way that both captures the covariate information and scales well with the covariate dimension; (2) the actual FDP is not directly available from the data; (3) direct optimization of (1) may cause overfitting and hence lose FDR control.

For the first challenge, intuitively, the decision threshold should have large values where the alternative hypotheses are enriched. Such enrichment pattern, as discussed in the `NeuralFDR` paper[12], usually consists of local "bumps" at certain covariate locations and a global "slope" that represents generic monotonic relationships. For example, the distance from TSS and the AAF in Fig. 2c correspond to the bump structure (at 0 and 0.5 respectively) whereas the rest of the covariates correspond to the slope structure. `AdaFDR` addresses these two structures by using a mixture of GLM and $K$-component Gaussian mixture (with diagonal covariance matrices), i.e.,

$$t(\mathbf{x}) = \exp(\mathbf{a}^T \mathbf{x} + b) + \sum_{k=1}^{K} \exp\left[w_k - (\mathbf{x} - \boldsymbol{\mu}_k)^T \mathrm{diag}(\boldsymbol{\sigma}_k)(\mathbf{x} - \boldsymbol{\mu}_k)\right], \tag{2}$$

where $\mathrm{diag}(\boldsymbol{\sigma}_k)$ represents a diagonal matrix with diagonal elements specified by the $d$-dimensional vector $\boldsymbol{\sigma}_k$. The set of parameters to optimize can be written as $\{\mathbf{a} \in \mathbb{R}^d, b \in \mathbb{R}, \{w_k \in \mathbb{R}, \boldsymbol{\mu}_k \in \mathbb{R}^d, \boldsymbol{\sigma}_k \in \mathbb{R}^d\}_{k=1}^{K}\}$. We choose to use the diagonal covariance matrices for Gaussian mixture to speed up the optimization. As a result, the number of parameters grows linearly with respect to the covariate dimension $d$, and the parameters can be easily initialized via an EM algorithm, as described below.

For the second challenge, we use a "mirror estimator" to estimate the number of FD of a given threshold function $t$,

$$\text{mirror estimator}: \widehat{\mathrm{FD}}(t) \overset{\text{def}}{=} \sum_{i=1}^{N} \mathbb{I}_{\{P_i \ge 1 - t(\mathbf{x}_i)\}}.$$

Such estimator has been used in recent works[12,15,49,50] and yields a conservative estimate of the true number of FD, in the sense that its expected value is larger than that of the true FD under mild assumptions (Lemma 1 in Supplementary Materials). Furthermore, FDP can be simply estimated as $\widehat{\mathrm{FDP}}(t) = \frac{\widehat{\mathrm{FD}}(t)}{\mathrm{D}(t)}$.

For the third challenge, `AdaFDR` controls FDP with high probability via hypotheses splitting. The hypotheses are randomly split into two folds; a separate decision threshold is learned on each fold and applied on the other. Since the learned threshold does not depend on the fold of data onto which it is applied, FDP can be controlled with high probability—such a statement is made formal in Theorem 1. We note that in multiple testing by `AdaFDR`, the learning-and-testing process is repeated twice, with each fold being the training set at one time and the testing set at the other. Figure 5 shows one of such process with fold 1 being the training set.

The full algorithm is described in Algorithm 1. Here, for example, $\mathrm{D}_{\mathrm{train}}(t)$, $\mathrm{D}_{\mathrm{test}}(t)$ are understood as the number of discoveries on the training set and the testing set respectively. Similar notations are used for other quantities like $\mathrm{FDP}(t)$ and the mirror estimate $\widehat{\mathrm{FDP}}(t)$ without explicit definition.

`AdaFDR` follows a similar strategy as our preliminary work `NeuralFDR`[12], which it subsumes: both methods use the mirror estimator to estimate FDP and use hypotheses splitting for FDP control. The main difference is on the modeling of the decision threshold $t$: `NeuralFDR` uses a neural network, which is flexible enough but hard to optimize. `AdaFDR`, in contrast, adopts the simpler mixture model that may lack certain flexibility but is much easier to optimize. This change of modeling, however, does not seems to reduce much of the detection power for `AdaFDR`. As shown in Supplementary Fig. 8b, the performance of `AdaFDR` is similar to that of `NeuralFDR`, while `AdaFDR` is orders of magnitude faster.

Algorithm 1
`AdaFDR` for multiple hypothesis testing
1: Randomly split the data $\mathcal{D} = \{(P_i, \mathbf{x}_i)\}_{i=1}^N$ into two folds $\mathcal{D} = \mathcal{D}_1 \cup \mathcal{D}_2$ of equal size.
2: **for** $(j, j') = (1, 2), (2, 1)$ **do**
3: Set $\mathcal{D}_j$ to be the training set and $\mathcal{D}_{j'}$ the testing set.
4: Learn the decision threshold $t^*(\mathbf{x})$ on the training set by optimizing

$$\text{maximize } \mathrm{D}_{\mathrm{train}}(t)\ s.t.\ \widehat{\mathrm{FDP}}_{\mathrm{train}}(t) \le \alpha. \tag{3}$$

5: Compute the best rescale factor $\gamma^*$ on the testing set

$$\gamma^* = \sup_{\gamma > 0}\{\gamma : \widehat{\mathrm{FDP}}_{\mathrm{test}}(\gamma t^*) \le \alpha\}. \tag{4}$$

6: Reject the hypotheses $\mathcal{R}_{j'} = \{i : i \in \mathcal{D}_{j'}, P_i \le \gamma^* t^*(\mathbf{x}_i)\}$.
7: Report discoveries on both folds $\mathcal{R} = \mathcal{R}_1 \cup \mathcal{R}_2$.

**Optimization.** Recall that the optimization is done solely on the training set $\mathcal{D}_{\mathrm{train}}$. Substituting FDP in Eq. (1) with its mirror estimate, we can rewrite the optimization problem as

$$\text{maximize}_{t \in \mathcal{T}} \mathrm{D}_{\mathrm{train}}(t),\ s.t.\ \frac{\widehat{\mathrm{FD}}_{\mathrm{train}}(t)}{\mathrm{D}_{\mathrm{train}}(t)} \le \alpha, \tag{5}$$

where $\mathcal{T}$, the set of decision thresholds to optimize over, corresponds to the mixture model (2). Our strategy is to first compute a good initialization point and then perform optimization by gradient descent on a relaxed problem. We note that a better solution to the optimization problem will give a better detection power. However, the FDP control guarantee holds *regardless* of the decision threshold we come up with.

- **Initialization:** Let $\pi_0(\mathbf{x})$ and $\pi_1(\mathbf{x})$ be the covariate distribution for the null hypotheses and the alternative hypotheses respectively. Following the intuition that the threshold $t(\mathbf{x})$ should be large when the number of alternative hypotheses is high and the number of null hypotheses is low, it is a good heuristic to let

$$t(\mathbf{x}) \propto \frac{\pi_1(\mathbf{x})}{\pi_0(\mathbf{x})}.$$

This is done in `AdaFDR` as follows. First, covariates with *p*-values larger than 0.75, i.e., $\{\mathbf{x}_i : i \in \mathcal{D}_{\mathrm{train}}, P_i \ge 0.75\}$, are treated as an approximate ensemble of the null hypotheses, and those with *p*-values smaller than the BH threshold, i.e., $\{\mathbf{x}_i : i \in \mathcal{D}_{\mathrm{train}}, P_i \le t_{\mathrm{BH}}\}$, are treated as an approximate ensemble of the alternative hypotheses. Then first, a mixture model same as Eq. (2) is fitted on the null ensemble $\{\mathbf{x}_i : i \in \mathcal{D}_{\mathrm{train}}, P_i \ge 0.75\}$ using an EM algorithm, resulting in an estimate of the null hypothesis distribution $\hat{\pi}_0(\mathbf{x})$. Second, each point in the alternative ensemble $\{\mathbf{x}_i : i \in \mathcal{D}_{\mathrm{train}}, P_i \le t_{\mathrm{BH}}\}$ receives a sample weight $1/\hat{\pi}_0(\mathbf{x})$. Last, the mixture model (2) is fitted on the weighted alternative ensemble using an EM algorithm to obtain the final initialization threshold. The details of the EM algorithm can be found in

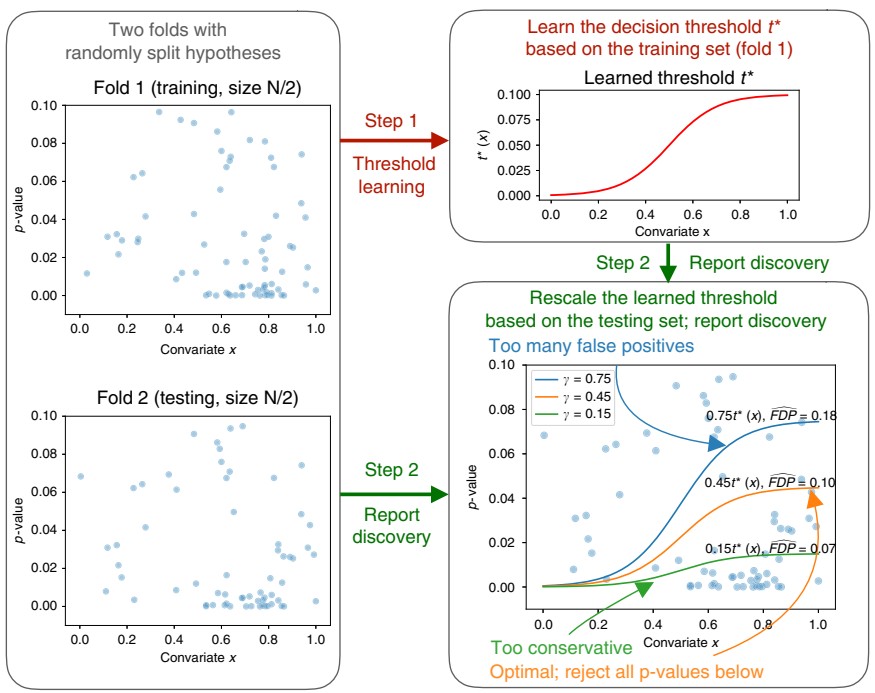

**Fig. 5** Schematic of the `AdaFDR` learning and testing process. Fold 1 is the training set and fold 2 is the testing set (left panel). In step 1, a decision threshold $t^*(\mathbf{x})$ is learned on the training set via solving the optimization problem (1) (upper-right panel). In step 2, as shown in the bottom-right panel, this learned threshold $t^*(\mathbf{x})$ is first rescaled by a factor $\gamma^*$, defined as the largest number whose corresponding mirror-estimated FDP on the testing set is less than $\alpha$ (orange). Then all $p$-values on the testing set below the rescaled threshold are rejected. Here the nominal FDP is $\alpha = 0.1$

Supplementary Note 1.3.

- **Optimization:** First, a Lagrangian multiplier is used to deal with the constraint:

$$\text{minimize}_{t \in \mathcal{T}} - D_{\text{train}}(t) + \lambda_1 [\widehat{\text{FD}}_{\text{train}}(t) - \alpha D_{\text{train}}(t)] \vee 0, \quad (6)$$

where $\lambda_1$ is chosen heuristically to be $10/\alpha$. Second, the sigmoid function is used to deal with the discontinuity of the indicator functions in $D_{\text{train}}(t)$ and $\widehat{\text{FD}}_{\text{train}}(t)$:

$$D_{\text{train}}(t) = \sum_{i \in \mathcal{D}_{\text{train}}} \mathbb{I}_{\{P_i \leq t(\mathbf{x}_i)\}} \approx \sum_{i \in \mathcal{D}_{\text{train}}} S[\lambda_0(t(\mathbf{x}_i) - P_i)],$$

$$\widehat{\text{FD}}_{\text{train}}(t) = \sum_{i \in \mathcal{D}_{\text{train}}} \mathbb{I}_{\{P_i \geq 1 - t(\mathbf{x}_i)\}} \approx \sum_{i \in \mathcal{D}_{\text{train}}} S[\lambda_0(P_i - 1 + t(\mathbf{x}_i))],$$

where $S(\cdot) = \frac{1}{1+e^{-x}}$ is the sigmoid function and $\lambda_0$ is automatically chosen at the beginning of the optimization such that the smoothed versions are good approximations to the original ones. Finally, the Adam optimizer[51] is used for gradient descent.

**FDP control**. We would like to point out that the mirror estimate is more accurate when its value is large. Hence, when the number of rejections is small ($<100$), the result should be treated with precaution. However, this should not be a major concern since in the target applications of `AdaFDR`, usually thousands to millions of hypotheses are tested simultaneously, and hundreds of hypotheses are rejected. In those cases, the mirror estimate is accurate. Hence for the theoretical result, we further require that for each fold, the best scale factor $\gamma^*$ should have a number of discoveries exceeding $c_0 N$ for some pre-specified small proportion $c_0$; failing to satisfy this condition will result in no rejection in this fold. In other words, we consider a modified version of Algorithm 1 with Eq. (4) substituted by setting

$$\gamma^* = \sup_{\gamma \geq 0} \{\gamma : \widehat{\text{FDP}}_{\text{test}}(\gamma t^*) \leq \alpha, D_{\text{test}}(\gamma t^*) \geq c_0 N\} \cup \{0\}. \quad (7)$$

Our FDP control on this modified version can be stated as follows.

**Theorem 1.** *(FDP control) Assume that all null $p$-values $P_i \in \mathcal{H}_0$, conditional on the covariates, are independently and identically distributed (i.i.d.) following Unif[0, 1]. Then with probability at least 1-$\delta$, AdaFDR with the modification (7) controls FDP at level $(1+\epsilon)\alpha$, where $\epsilon = O\left(\sqrt{\frac{\log\frac{1}{\delta}}{\alpha N}}\right)$.*

The assumption made in Theorem 1 is standard in the literature[15,17] and can be easily relaxed to the assumption that the null $p$-values, conditional on the covariates, are independently distributed and stochastically greater than Unif[0, 1]

(Supplementary Note 3.1). In addition, Theorem 1 is strictly stronger than the one for `NeuralFDR` (Supplementary Note 1.2).

**Covariate visualization via `AdaFDR_explore`.** `AdaFDR` also provides a Feature-Explore function that can visualize the relationship between each covariate and the significance of hypotheses, in terms of the marginal covariate distribution for the null hypotheses and the alternative hypotheses respectively, as those shown in Figs. 2c and 3b–e. Let $x_i$ be the univariate covariate under consideration and $h_i = 0/1$ indicate the ground truth (true null/alternative) for the $i$th hypothesis. Then here we are trying to estimate the conditional covariate distribution given the hypothesis label, i.e., $\mathbb{P}(x_i|h=0)$ and $\mathbb{P}(x_i|h=1)$. Noting that as a function of $x_i$,

$$\frac{\mathbb{P}(x_i|h=1)}{\mathbb{P}(x_i|h=0)} \propto \frac{\mathbb{P}(x_i, h=1)}{\mathbb{P}(x_i, h=0)} = \frac{\mathbb{P}(h=1|x_i)}{\mathbb{P}(h=0|x_i)}.$$

The ratio between the two distributions can also be interpreted, up to a scale factor, as the ratio of the hypothesis being true alternative/null given the covariate.

The estimation is done as follows. First, for the entire dataset, covariates with $p$-values greater than 0.75, i.e., $\{x_i : i \in [N], P_i \geq 0.75\}$, are treated as an approximate ensemble of the null hypotheses, and those with $p$-values less than the BH threshold, i.e. $\{x_i : i \in [N], P_i \leq t_{\text{BH}}\}$, are treated as an approximate ensemble of the alternative hypotheses. Then for each covariate, the null hypothesis distribution and the alternative hypothesis distribution are estimated from these two ensembles using kernel density estimation (KDE) for continuous covariates and simple count estimator for categorical covariates. In addition, for categorical covariates, the categories are reordered based on the ratio between the estimated alternative probability and null probability $\hat{\pi}_1(\mathbf{x})/\hat{\pi}_0(\mathbf{x})$.

**Experiment details**. The default parameters of `AdaFDR` are used for every experiment in this paper, both real data analysis and simulations, without any tuning. Specifically, the number of Gaussian mixture components $K$ is fixed to be 5, and the number of iterations for the optimization step is fixed to be 1500. We found that the performance is not sensitive to these parameter choices. For input, `AdaFDR` also allows filtered data with only small $p$-values close to 0 and large $p$-values close to 1, which could accelerate the algorithm. For the GTEx data, the data are filtered to have only data points with $p$-values $P_i < 0.01$ or $P_i > 0.99$. In such a case, the original number of hypotheses (before filtering) is required as input to control FDR. Details about this can be found in Supplementary Note 1.5. Details of other methods can be found in Supplementary Note 1.6. Step-by-step documentation for most experiments can be found on GitHub (https://github.com/martinjzhang/AdaFDRpaper).

**Reporting summary**. Further information on research design is available in the Nature Research Reporting Summary linked to this article.

## Data availability

Most of the data, including the curated data for GTEx and all other applications, are available at online data repository (https://osf.io/6krzn/) with downloading instructions available on GitHub (https://github.com/martinjzhang/AdaFDRpaper). All other relevant data are available upon request.

## Code availability

We have released a Python implementation of AdaFDR on GitHub (https://github.com/martinjzhang/adafdr) and an R wrapper on GitHub (https://github.com/fxia22/RadaFDR). The Python package is also available at PyPI with the name adafdr. The software version adafdr 0.1.7 is used for all experiments. The code and the documentation for reproducing most of the results of the paper is available on GitHub (https://github.com/martinjzhang/AdaFDRpaper).

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

## Acknowledgements

We would like to thank David Tse, Liuhua Lei, Nikolaos Ignatiadis and Vivek Bagaria for helpful discussions. M.Z. and F.X. are partially supported by Stanford Graduate Fellowship. M.Z. is partially supported by the Stanford Data Science Initiative, NSF Grant CCF 0939370 (Center for Science of Information), NSF Grant CIF 1563098, and NIH Grant R01HG008164. J.Z. is supported by the Chan-Zuckerberg Initiative and National Science Foundation (NSF) Grant CRII 1657155.

## Author contributions

M.Z. and F.X. designed the algorithm and conducted the experiments. M.Z. performed the theoretical analysis. M.Z. and J.Z. wrote the manuscript. J.Z. supervised the research. All authors reviewed the manuscript.

## Additional information

**Competing interests:** The authors declare no competing interests.

