## [Peer Review File · Nature Communications]

Reviewers' comments:

Reviewer #1 (Remarks to the Author):

Zhang et al. describe AdaFDR, a fast procedure to select covariate-specific p-value cutoffs at a predefined FDR. The procedure post-processes p-values based on a linear regression, then model the relationships between the p-values and covariates. The authors showed that AdaFDR outperformed a set of methods independent hypothesis weighting (IHW) and NeuroIFDR in multiple datasets. The procedure is promising.

(1) There are multiple ways to incorporate covariates into association or regression models, such as Bayesian hierarchical model (Gaffney et al. 2012). It is important to compare AdaFDR with hierarchical models in terms of detecting eQTLs.

(2) The authors compared AdaFDR's results with SBH's. However, whether the AdaFDR only associations are biological meaningful is not clear.

Reviewer #2 (Remarks to the Author):

The authors proposed a multiple testing procedure with integration of additional covariates. The manuscript is well written. The method is illustrated on selected tissues from GTEx data and simulation studies are conducted as well. However, it is unclear why a biased multiple testing procedure would be preferred over an unbiased one. And how one can use key covariates and annotations learned from the data to make prioritize on new discovery. There are also many other major issues listed below.

1. On the one hand, some of that information may help prioritize true signals in certain data sets, and on the other hand, they may bias the analysis and miss more opportunities to make new discoveries with unknown covariates.
2. What if the covariates are not informative? Would they hurt the analysis?
3. The authors are correct that the problem of multiple testing with covariates or other information has been actively explored. In addition to the references cited, there are other recent developments, for example, "a unified treatment of multiple testing with prior knowledge." arXiv:1703.06222. More comparison with methods allowing for covariates would be helpful.
4. A higher number of discoveries and a higher number of replicated eQTLs are illustrative but not enough. In one particular data set, maybe the replicated eQTLs are enriched with associations to one of the covariates. A more comprehensive evaluation and replication of all kinds of annotations and covariates on multiple data sets with replication would better convince the reviewer. Especially the discovery results on the SNPs that are not being prioritized by the covariates, how much we are going to lose by incorporating covariates.
5. In addition to multiple testing procedures, there are many joint analysis methods or integrative analysis methods may better serve the purpose of the data analysis. For example, colocalization analysis of eQTLs and other SNP annotations (PMID: 28278150).
6. The dependence among SNPs under the null (LD) is a known challenge, especially when LD is strong or even perfect. Some claims made in the manuscript need to be revisited to "...be extended to allow arbitrary dependency".

Reviewer #3 (Remarks to the Author):

The authors introduce a new statistical method, AdaFDR, for controlling the false discovery rate (FDR) while performing multiple hypothesis testing. AdaFDR joins several other methods developed over the last 5 years for controlling FDR while making use of "side information" or "covariates" (aside from just the p-values) to increase power over traditional FDR-controlling methods such as Benjamini and Hochberg's step-up procedure (BH) and the q-value. AdaFDR is shown to be computationally manageable, while also providing substantial gains over other methods in total discoveries (assumed to be true positives) across several computational biology problems (eQTL analysis, RNA-seq differential analysis, microbiome association analysis, and others). The gains of AdaFDR may be attributed to [1] the method's ability to make use of multiple informative covariates simultaneously and [2] the use of a flexible class of GLM and gaussian mixture models to estimate the decision boundary at a specified nominal FDR threshold (e.g. 0.05).

The vignettes and simulations presented show impressive performance, and the method could prove to be a great tool for increasing power when performing multiple testing correction. However, I have several concerns which need to be addressed before the paper is suitable for publication, regarding [1] the methods and covariates used in the comparisons, [2] the stability of the method, [3] the limited usability of the software, and [4] the interpretation of the results. Details are provided as bullet points below.

Regarding comparison.

1. In the main analysis described in the Results section (eQTLs in GTEx), AdaFDR and AdaFDR-fast are only compared against BH, q-value, and in some cases IHW. While the size of the problem may make the comparison of some methods (e.g. AdaPT) computationally unreasonable, other notable methods are missing from the analysis. Several covariate-aware methods are referenced in the paper (bottom of page 2, refs 32-38). It is unclear why most of these methods were included in the comparison. This needs to be clarified, and at the very least, the method of Boca and Leek (2018) should be included in the comparison. The Boca and Leek (2018) approach is similarly able to handle multiple covariates and seems like a reasonable and meaningful comparison.
2. The authors state: "the standard assumption of AdaFDR and all the related methods is that the covariates should not affect the p-values under the null hypothesis" (page 2). Have the authors verified that the covariates used in the data analyses are indeed independent of the p-value under the null? I am particularly concerned that the independence assumption is violated when the p-values from a separate eQTL analysis are used as the covariate. This needs to be checked and demonstrated since violation of the assumption can lead to significant false positives.
3. The "AdaFDR" and "AdaFDR-fast" methods are labelled as just "AdaFDR" in Figures 3. This is confusing and unclear. While I recognized that both are contributions of the paper, from a practical perspective, for a **user**, these two are not the same as they require setting different parameters. At the very least, Figure 3 and the accompanying legend should be updated to clarify when results for AdaFDR or AdaFDR-fast are reported (to match other results tables, e.g. Figure 2).
4. Building on the last point, AdaFDR-fast (rather than AdaFDR) is used in both the Microbiome and Proteomics data sets "due to the small sample size." This distinction of when to use AdaFDR-fast or AdaFDR (and the definition of "small sample size") needs to be made clear in the Discussion or Introduction.

Regarding stability.

5. Based on Algorithm 1 (page 7), the AdaFDR method appears to estimate a decision threshold based on splitting the data into two (random) folds. How sensitive is the method to the random split, i.e.

how different are the resulting significance calls if the hypotheses are reordered? I can imagine a scientist using the method, removing one or two tests or reordering their p-values and re-running the method and being surprised at getting different sets of significant calls. How much will the calls change if the data sets analyzed in the Results section were reordered?

6. AdaFDR estimates a threshold for a specified nominal FDR threshold (α). Presumably, the method needs to be re-run every time a different α threshold is wanted, e.g. to obtain approximate "q-values". While the flexibility of the GLM-gaussian mixture model allows for estimating complex decision boundaries, I am curious whether this also poses challenges for monotonicity. That is, do decision boundaries frequently cross such that a hypothesis is significant for some α , but not significant for some $\alpha^* > \alpha$? Does this occur for the data sets analyzed when comparing common α cutoffs (e.g. 0.01, ..., 0.10)? Again, this may cause surprise for anyone using the method.

Regarding usability.

7. The authors have done a good job in making the AdaFDR software (github.com/martinjzhang/adafdr) and the analysis performed for the paper (github.com/martinjzhang/AdaFDRpaper) available on GitHub and pypi. However, I am concerned with the limited usability of the method. Large scale statistical inference (i.e. hypothesis testing) in genomics and bioinformatics is often performed in R or with command line tools and not Python (note also that all other FDR controlling methods benchmarked in this paper are also implemented in R). The current software could be greatly improved if the authors provided an example for calling the AdaFDR Python function from R, e.g. using the "reticulate" package (<https://rstudio.github.io/reticulate/>) to access a wider audience of users.

Regarding interpretation.

8. Precise language needs to be used to describe the results. The descriptions and labels of the null and alternative hypothesis distribution plots (e.g. Figure 2C) are misleading. The plots show $f(x \mid \text{pval not significant})$ and $f(x \mid \text{pval significant})$, the conditional distributions of the covariate given that a test shows either weak (large p-value) or strong (small p-value) evidence for being significant. However, the plots are labeled as "null/alternative proportion", which suggests something else - namely, $p(h = 0 \mid x)$ and $p(h = 1 \mid x)$. This is made even more confusing as the discussion of these plots in the text make statements of $p(\text{pval significant} \mid x)$ and $p(\text{pval not significant} \mid x)$, e.g. "genes with higher expression levels are more likely to have significant associations" (page 3). First, and most importantly, "significant associations" should not be used interchangeably with tests being truly alternative (as in the plot label), This is not correct. Second, even if the two were the same, the conditioning is being flipped between the text/labels and the plots - further confusing the interpretation of these results. These details need to be clarified as it impacts the interpretation of the results presented in the paper. For example, while "genes with higher expression levels are more likely to have significant associations" (page 3), this **should not** be interpreted as genes having higher expression levels being more likely to be truly differential, as currently implied by Figure 2C and stated on page 4: "alternative hypotheses are more likely to occur when the expression levels are high." This is not a correct interpretation of the data. While more significant p-values occur when the expression levels are high, this can be due to reasons other than the association between alternative hypotheses and expression levels claimed in the text. Instead, it may simply be due to better power to detect differences when the expression levels are high.

Minor Issues

- In general, the Introduction is awkwardly organized, with (somewhat redundant) subsections and a

few mislabeled references. This should be cleaned up.

- Throughout, "multiple hypothesis testing" is used to mean "multiple testing correction". This should be corrected.
- On page 2, mathematical notation is introduced without ever being mentioned again in the main text and can be dropped.
- The descriptions of the various applications (e.g. RNA-seq differential expression analysis) should be made more precise. For example, "RNA-seq data" is used synonymously with differential gene expression analysis with RNA-seq data (page 2).

I agree to have my name released.

Patrick Kosuke Kimes, PhD
Postdoctoral Research Fellow
Department of Data Sciences, Dana-Farber Cancer Institute
Department of Biostatistics, Harvard TH Chan School of Public Health

We thank the reviewers for their thoughtful feedback. We provide point-by-point response to all of the reviewers' questions below.

REVIEWER 1 COMMENTS:

Zhang et al. describe AdaFDR, a fast procedure to select covariate-specific p-value cutoffs at a predefined FDR. The procedure post-processes p-values based on a linear regression, then model the relationships between the p-values and covariates. The authors showed that AdaFDR outperformed a set of methods independent hypothesis weighting (IHW) and NeuroFDR in multiple datasets. The procedure is promising.

Thank you for your careful review and helpful suggestions.

(1) There are multiple ways to incorporate covariates into association or regression models, such as Bayesian hierarchical model (Gaffney et al. 2012). It is important to compare AdaFDR with hierarchical models in terms of detecting eQTLs.

Thank you for pointing us to this paper. In the context of eQTL studies, the mentioned work [Gaffney, et al., 2012] presents a post-hoc analysis that, given the eQTL discoveries, prioritize the causal SNPs using regulatory annotations. This is different from our work because 1) our work directly incorporates the covariates into the discovery process while the mentioned work uses annotations for the post-hoc analysis; 2) our work considers general covariates and provides FDR control guarantee while the mentioned work specifically considers regulatory annotations without guarantees regarding the false positives. We will clarify this in the revision as below.

of 0.01. Such experiment of testing all SNP-gene pairs simultaneously is a prescreening step for detecting casual eQTLs and is also performed in some recent works^{12,13}. A similar analysis workflow is to first discover significant genes (eGenes) and then match significant SNPs (eVariants) for each eGene⁴². There are also works that, given the eQTL discoveries, prioritize the casual SNPs based on regulatory annotations in a post-hoc fashion⁴⁴ or use eQTL findings to help identify casual SNPs in GWAS⁴⁷.

Also, since the focus of the present paper is on developing a general statistical method instead of eQTL study, we compared our method AdaFDR with other general statistical methods (AdaPT [Lei, et al., 2018], IHW [Ignatiadis et al, 2016], BL [Boca and Leek, 2018]) instead of methods specifically developed for eQTL studies. Moreover, we have demonstrated how AdaFDR improves discovery in several other biological settings such as RNA-seq, microbiome, proteomics, and fMRI data.

(2) The authors compared AdaFDR's results with SBH's. However, whether the AdaFDR only associations are biological meaningful is not clear.

Thank you for the comment. To further investigate the biological meanings of the AdaFDR-only discoveries, we plotted the marginal distribution of AdaFDR-only discoveries and SBH-only discoveries over each biological covariate in Supplementary Figure 2, copied below. The results for different tissues are similar, so we only included the results for Adipose_Subcutaneous and Colon_Sigmoid.

As shown in the figure below, there is a higher proportion of AdaFDR-only discoveries at locations where 1) the distance from TSS is small (upper left); 2) the SNP has an active chromatin state (upper right); 3) the SNP AAF is close to 0.5 (lower left); 4) the gene expression level is neither too high or too low (lower right). All these match the enrichment pattern of eQTLs, indicating that AdaFDR-only discoveries are more biologically relevant. We have added this as a supplementary figure in the revision.

As part of the validation, we have also shown that the AdaFDR-only discoveries have much smaller p-values than SBH-only discoveries on an independent eQTL dataset (MuTHER) of the same tissue. Previously we have done it for the tissues Adipose_Subcutaneous and Adipose_Visceral_Omentum. As shown in the figure below (right panel), we added a third validation on the tissue Cells_EBV-transformed_lymphocytes, where we observe a similar result that AdaFDR-only discoveries have much smaller p-values on the independent MuTHER dataset.

Since the paper focuses on developing a general statistical method instead of eQTL studies, we do not include more downstream validations of the eQTL discoveries.

C validation on MuTHER

Supplementary Figure 1. Additional results on the GTEx data. (a) Results on the two colon tissues. (b) Feature visualization for Colon_Sigmoid (c) Validation for GTEx Adipose_Visceral_Omentum using the MuTHER adipose eQTL data (left) and for GTEx Cells_EBV-transformed_lymphocytes using the MuTHER lymphocytes (LCL) eQTL data (right).

REVIEWER 2 COMMENTS:

The authors proposed a multiple testing procedure with integration of additional covariates. The manuscript is well written. The method is illustrated on selected tissues from GTEx data and simulation studies are conducted as well. However, it is unclear why a biased multiple testing procedure would be preferred over an unbiased one. And how one can use key covariates and annotations learned from the data to make prioritize on new discovery. There are also many other major issues listed below.

Thank you for your review and helpful comments. We want to clarify that AdaFDR has no prior bias over how to prioritize hypothesis based on covariates. It learns everything entirely in a data-driven manner, similar to popular state-of-the-art methods such as IHW [Ignatiadis et al, 2016], AdaPT [Lei, et al., 2018], and BL [Boca and Leek, 2018]. All of our experiments demonstrate that AdaFDR makes substantially more discoveries than methods that do not use side information, while controlling FDR. We provide a point-to-point response as below.

1. On the one hand, some of that information may help prioritize true signals in certain data sets, and on the other hand, they may bias the analysis and miss more opportunities to make new discoveries with unknown covariates.

Thank you for the comment. First, we do not consider AdaFDR as a biased multiple testing procedure since it does not assume any prior knowledge about the covariates; it learns the relationship between the covariates and the p-values in an unbiased data-driven manner. Second, as shown in the figure (panel a) below, the proportion of SBH-only discoveries is tiny for all 17 tissues, indicating that AdaFDR would not miss many discoveries made by the non-adaptive methods (SBH here). We have added this as a supplementary figure in the revision. The current state-of-the-art methods such as IHW, AdaPT and BL also use side information to prioritize hypothesis in a similar data-driven manner, and they are not considered to be biased.

Supplementary Figure 2. (a) Result comparison between SBH and AdaFDR. For all 17 GTEx tissues, AdaFDR missed a tiny proportion of SBH discoveries while having substantially more other discoveries. (b-c) The

2. What if the covariates are not informative? Would they hurt the analysis?

Thank you for the comment. When the covariate is not informative, AdaFDR will have similar performance as the non-adaptive method SBH. We have mentioned this in the paper (simulation studies, page 5):

AdaFDR achieves greater power than all other methods while controlling FDR (Supplementary Figure 9, 10). AdaFDR reduces to SBH when the covariate is not informative, indicating that it is not overfitting the uninformative covariate (Supplementary Figure 9e).

The corresponding figure is shown as below (Supp. Fig. 9e):

3. The authors are correct that the problem of multiple testing with covariates or other information has been actively explored. In addition to the references cited, there are other recent developments, for example, “a unified treatment of multiple testing with prior knowledge.” arXiv:1703.06222. More comparison with methods allowing for covariates would be helpful.

Thanks for pointing us to this work. The mentioned work [Ramdas et al., 2017] considers structured covariates (covariate-dependent null proportion, hypothesis weights, grouping information, etc) while our work considers general covariates without prior information about their connection to the p-values. Therefore, the two methods are not directly comparable. We have cited this work among other statistical literatures in page 2 of the revision.

We have compared AdaFDR with AdaPT [Lei, et al., 2018], IHW [Ignatiadis et al, 2016], and BL [Boca and Leek, 2018], the three methods that are recommended in a recent comparison paper [Korthauer et al, 2018]. In the previous version, the performance of BL was only reported for a subset of simulation studies (Supp. Figures 9-10). We have added BL to all other experiments except the eQTL study (due to computational concerns) in the current version, namely those in Figures 3-4 and Supp. Figure 8. Overall, the additional results agree with the previous results (Supp. Figures 9-10): BL controls FDR but has less power than AdaFDR; its running speed is slower than AdaFDR-fast (Figure 4b).

4. A higher number of discoveries and a higher number of replicated eQTLs are illustrative but not enough. In one particular data set, maybe the replicated eQTLs are enriched with associations to one of the covariates. A more comprehensive evaluation and replication of all kinds of annotations and covariates on multiple data sets with replication would better convince the reviewer. Especially the discovery results on the SNPs that are not being prioritized by the covariates, how much we are going to lose by incorporating covariates.

Thank you for the comment. To answer these questions, we performed more analysis on the GTEx data as below, which we have added as supplementary figures in the revision.

First, to investigate the contribution of each covariate, we run AdaFDR using each covariate separately for all 17 tissues as below. The distance from TSS is the most informative while other covariates have smaller but still notable effects. Also, the combined improvement of using all covariates (31.9%) is similar to the sum of the four individual improvements (33.0%), indicating that the four covariates carry very different information regarding the hypotheses.

Second for the validation, previously we have shown that the AdaFDR-only p-values are much smaller than SBH-only p-values on an independent eQTL dataset (MuTHER) for the tissues Adipose_Subcutaneous and Adipose_Visceral_Omentum. We added a third validation on the tissue Cells_EBV-transformed_lymphocytes (right panel below), where we observe a similar result that AdaFDR-only discoveries have much smaller p-values on the independent MuTHER dataset.

Third, we compared the AdaFDR-only discoveries with SBH-only discoveries for all 17 tissues in the figure below. We found that the proportion of SBH-only discoveries is tiny, indicating that AdaFDR would not miss many discoveries made by the non-adaptive methods (SBH here).

Supplementary Figure 2. (a) Result comparison between SBH and AdaFDR. For all 17 GTEx tissues, AdaFDR missed a tiny proportion of SBH discoveries while having substantially more other discoveries. (b-c) The

5. In addition to multiple testing procedures, there are many joint analysis methods or integrative analysis methods may better serve the purpose of the data analysis. For example, colocalization analysis of eQTLs and other SNP annotations (PMID: 28278150).s

Thank you for pointing us to this paper. The mentioned work [Wen, et al., 2017] uses eQTL discoveries as annotations to help identify causal SNPs in GWAS. This is different from our work because 1) it uses eQTL discoveries as annotations for GWAS analysis, and the eQTL analysis itself does not use additional annotations; 2) it considers the specific case of using eQTL discoveries as the covariate without guarantee on false positives, while our work considers general covariates and provides FDR control guarantee. We will mention this in the revision as below.

of 0.01. Such experiment of testing all SNP-gene pairs simultaneously is a prescreening step for detecting casual eQTLs and is also performed in some recent works^{12,13}. A similar analysis workflow is to first discover significant genes (eGenes) and then match significant SNPs (eVariants) for each eGene⁴². There are also works that, given the eQTL discoveries, prioritize the casual SNPs based on regulatory annotations in a post-hoc fashion⁴⁴ or use eQTL findings to help identify casual SNPs in GWAS⁴⁷.

Also, since the focus of the present paper is on developing a general statistical method instead of eQTL study, we compared our method AdaFDR with other general statistical methods (AdaPT [Lei, et al., 2018], IHW [Ignatiadis et al, 2016], BL [Boca and Leek, 2018]) instead of methods specifically developed for eQTL studies. Moreover, we have demonstrated how AdaFDR improves discovery in several other biological settings such as RNA-seq, microbiome, proteomics, and fMRI data.

6. The dependence among SNPs under the null (LD) is a known challenge, especially when LD is strong or even perfect. Some claims made in the manuscript need to be revisited to “...be extended to allow arbitrary dependency”.

Thank you for the comment. We have added a detailed description on extending AdaFDR to allow arbitrary dependency in the supplementary material.

2.4 Extension to dependent case

Here we describe a simple procedure that extends AdaFDR to allow arbitrary dependency of p-values, borrowing ideas from the extended version of IHW¹³. The procedure can be described as follows:

1. Partition the hypotheses into two folds that are independent of each other, i.e., $\{(P_i, \mathbf{x}_i)\}_{i \in \mathcal{D}_1}$ and $\{(P_i, \mathbf{x}_i)\}_{i \in \mathcal{D}_2}$ that are mutually independent.
2. For each fold $j = 1, 2$, let $\{t_i\}_{i \in \mathcal{D}_j}$ be the threshold learned from the other fold (up to a scaling factor). Weight the p-values by

$$\tilde{P}_i = P_i \frac{\sum_{i \in \mathcal{D}_j} t_i}{|\mathcal{D}_j| t_i},$$

where $|\mathcal{D}_j|$ is the cardinality of the set \mathcal{D}_j .

3. Apply the BH procedure on the set of weighted p-values $\{\tilde{P}_i\}_{i \in [N]}$ with nominal FDR level $\alpha / \sum_{i \in [N]} \frac{1}{t_i}$.

We note that in eQTL studies, SNPs from different chromosomes can be regarded as being independent of each other. Also, the third step corresponds to the Benjamini-Yekutieli procedure⁶. By Theorem 1 in¹³, the above procedure controls FDR under arbitrary dependency of p-values. More specifically, it controls FDR under the assumptions:

1. The two folds are independent of each other.
2. The null p-values, conditional on the covariates, are independent and stochastically greater than the uniform distribution.

As a side note, in practice, the dependent case will have minimum effect as long as the two folds are independent, i.e., step 1 in the procedure above.

REVIEWER 3 COMMENTS:

The authors introduce a new statistical method, AdaFDR, for controlling the false discovery rate (FDR) while performing multiple hypothesis testing. AdaFDR joins several other methods developed over the last 5 years for controlling FDR while making use of "side information" or "covariates" (aside from just the p-values) to increase power over traditional FDR-controlling methods such as Benjamini and Hochberg's step-up procedure (BH) and the q-value. AdaFDR is shown to be computationally manageable, while also providing substantial gains over other methods in total discoveries (assumed to be true positives) across several computational biology problems (eQTL analysis, RNA-seq differential analysis, microbiome association analysis, and others). The gains of AdaFDR may be attributed to [1] the method's ability to make use of multiple informative covariates simultaneously and [2] the use of a flexible class of GLM and gaussian mixture models to estimate the decision boundary at a specified nominal FDR threshold (e.g. 0.05).

The vignettes and simulations presented show impressive performance, and the method could prove to be a great tool for increasing power when performing multiple testing correction. However, I have several concerns which need to be addressed before the paper is suitable for publication, regarding [1] the methods and covariates used in the comparisons, [2] the stability of the method, [3] the limited usability of the software, and [4] the interpretation of the results. Details are provided as bullet points below.

Thank you for your very thoughtful review and helpful suggestions.

Regarding comparison.

1. In the main analysis described in the Results section (eQTLs in GTEx), AdaFDR and AdaFDR-fast are only compared against BH, q-value, and in some cases IHW. While the size of the problem may make the comparison of some methods (e.g. AdaPT) computationally unreasonable, other notable methods are missing from the analysis. Several covariate-aware methods are referenced in the paper (bottom of page 2, refs 32-38). It is unclear why most of these methods were included in the comparison. This needs to be clarified, and at the very least, the method of Boca and Leek (2018) should be included in the comparison. The Boca and Leek (2018) approach is similarly able to handle multiple covariates and seems like a reasonable and meaningful comparison.

Thank you for the comment. We chose AdaPT [Lei, et al., 2018], IHW [Ignatiadis et al, 2016], and BL [Boca and Leek, 2018] for comparison since they are the three methods that control FDR as evaluated in a recent comparison paper [Korthauer et al., 2018]. In the previous version, the performance of BL was only reported for a subset of simulation studies (Supp. Figures 9-10). We have added BL to all other experiments except the eQTL study in the current version, namely those in Figures 3-4 and Supp. Figure 8.

Overall, the additional results agree with the previous results (Supp. Figures 9-10): BL controls FDR but has less power than AdaFDR; its running speed is between AdaFDR-fast and AdaFDR (Figure 4b).

There are two reasons that we did not run BL on the full GTEx data (Figure 2). First, BL requires the complete data to be loaded into the memory which is too much for the full GTEx. Other methods circumvent this problem by either using p-value filtered data (e.g., containing only hypotheses with very small and large p-values) or learning using a subset of data and generating the covariate-adaptive p-value

weights for the rest of the data in a sequential manner. Second, BL did not show promising performance on the small GTEx experiments (Figure 3a). Since the small GTEx datasets are representative of the full GTEx data, it is reasonable to expect that BL will not yield a good result on the full GTEx data.

2. The authors state: "the standard assumption of AdaFDR and all the related methods is that the covariates should not affect the p-values under the null hypothesis" (page 2). Have the authors verified that the covariates used in the data analyses are indeed independent of the p-value under the null? I am particularly concerned that the independence assumption is violated when the p-values from a separate eQTL analysis are used as the covariate. This needs to be checked and demonstrated since violation of the assumption can lead to significant false positives.

Thank you for the comment. To verify the assumption for the GTEx experiments, we plotted the p-value histograms stratified by each covariate separately (see the figures below for Adipose_Subcutaneous and Colon_Sigmoid). We found that all histograms show a mixture of a uniform distribution and an enrichment of small p-values to the left, indicating that the null p-values are uniformly distributed independent of the covariate. This also includes the case where the p-value from a separate eQTL analysis (of the matching tissue) is used as the covariate. We have added the following two figures as supplementary figures in the revision. Similar diagnostic plots were also used in the IHW paper [Ignatiadis et al., 2016].

Supplementary Figure 4. To verify the algorithm assumption (Theorem 1) for the GTEx experiments, we plot the p-value histograms stratified by each covariate separately for the tissue Adipose_Subcutaneous. All histograms show a mixture of a uniform distribution and an enrichment of small p-values to the left, indicating that the null p-values are uniformly distributed independent of the covariate.

Supplementary Figure 5. P-value histograms stratified by each covariate separately for the tissue Colon_Sigmoid. Similar to Supplementary Figure 4.

3. The "AdaFDR" and "AdaFDR-fast" methods are labelled as just "AdaFDR" in Figures 3. This is confusing and unclear. While I recognized that both are contributions of the paper, from a practical perspective, for a ****user****, these two are not the same as they require setting different parameters. At the very least, Figure 3 and the accompanying legend should be updated to clarify when results for AdaFDR or AdaFDR-fast are reported (to match other results tables, e.g. Figure 2).

Thank you for the comment. We have added a clarification in the caption of Figure 3 as below.

Figure 3. (a) The number of discoveries of various methods on two small GTEx eQTL datasets, three RNA-Seq datasets, two microbiome datasets, one proteomics dataset, and two fMRI datasets. **AdaFDR is used for the small GTEx and the RNA-Seq datasets while AdaFDR-fast is used for others, due to their smaller data size.** The fMRI results for AdaPT are omitted since the AdaPT software does not support categorical covariates. (b) Covariate visualization for RNA-Seq datasets. (c) Covariate visualization for microbiome dataset. (d) Covariate visualization for proteomics dataset. (e) Covariate visualization for fMRI datasets.

4. Building on the last point, AdaFDR-fast (rather than AdaFDR) is used in both the Microbiome and Proteomics data sets "due to the small sample size." This distinction of when to use AdaFDR-fast or AdaFDR (and the definition of "small sample size") needs to be made clear in the Discussion or Introduction.

Thank you for your comment. We have added a clarification in Discussion as below.

The typical use-case for AdaFDR is when there are many hypotheses to be tested simultaneously — ideally more than 10k. This is because AdaFDR needs many data to learn the covariate-adaptive threshold and to have an accurate estimate of FDP. A similar recommendation on the number of hypotheses is also made for IHW. **When there are less hypotheses (<10k) or it is expected to have very few discoveries (less than a few hundreds), AdaFDR-fast is recommended as a more robust choice.** Also, when we have a smaller number of hypotheses, the discoveries are still valid but need to be treated with precaution — ideally with some orthogonal validations.

Regarding stability.

5. Based on Algorithm 1 (page 7), the AdaFDR method appears to estimate a decision threshold based on splitting the data into two (random) folds. How sensitive is the method to the random split, i.e. how different are the resulting significance calls if the hypotheses are reordered? I can imagine a scientist using the method, removing one or two tests or reordering their p-values and re-running the method and being surprised at getting different sets of significant calls. How much will the calls change if the data sets analyzed in the Results section were reordered?

Thank you for the comment. We agree that stability is a desirable property for the algorithm. To investigate the stability of AdaFDR, we reran all 10 experiments in Figure 3a (with the same setting and different random seeds) 50 times and found the number of discoveries to be highly consistent (first column in the figure below). Furthermore, for each of the 50 repetition, we run AdaFDR for a second time and find most discoveries can be reproduced (the average replication rate is 92.4% across the ten datasets). This shows good stability of the algorithm. The results are summarized in the following figure which is included as a supplementary figure.

	discoveries (std)	reproduced discoveries %
small_GTEx: Adipose_Subcutaneous	1491 (41)	94.5%
small_GTEx: Adipose_Visceral_Omentum	1396 (96)	89.5%
RNA-Seq: Bottomly	2147 (38)	93.6%
RNA-Seq: Pasilla	830 (15)	94.4%
RNA-Seq: airway	6041 (33)	97.2%
microbiome: enigma_ph	119 (8)	87.8%
microbiome: enigma_al	480 (46)	82.4%
proteomics	408 (18)	89.6%
fMRI: auditory	1066 (10)	96.9%
fMRI: imagination	2233 (12)	97.6%

Supplementary Figure 7. AdaFDR may produce slightly different results in different runs on the same dataset due to its inherent randomness. To showcase its stability, we repeat all 10 experiments in Figure 3a 50 times with different random seeds. As shown in the first column of the table, the number of discoveries of the 50 repetitions are highly consistent. Furthermore, for each of the 50 repetitions, we run AdaFDR for a second time and report the proportion of reproduced discoveries in the second column of the table (number of overlapped discoveries in both runs divided by average number of discoveries in the first run). The average replication rate is 92.4% across the ten datasets, indicating good stability of the algorithm. The two microbiome datasets have relatively lower replication rate (87.8% and 82.4%, respectively), due to their smaller data size (~ 4000 hypotheses).

6. AdaFDR estimates a threshold for a specified nominal FDR threshold (α). Presumably, the method needs to be re-run every time a different α threshold is wanted, e.g. to obtain approximate "q-values". While the flexibility of the GLM-gaussian mixture model allows for estimating complex decision boundaries, I am curious whether this also poses challenges for monotonicity. That is, do decision boundaries frequently cross such that a hypothesis is significant for some α , but not significant for some $\alpha^* > \alpha$? Does this occur for the data sets analyzed when comparing common α cutoffs (e.g. 0.01, ..., 0.10)? Again, this may cause surprise for anyone using the method.

Thank you for the comment. We have provided a new retest function *adafdr_retest* that, given the testing result from the main test function *adafdr_test*, produces the testing result for other different nominal FDR levels. Therefore, the user only needs to run AdaFDR once with *adafdr_test*. And whenever he/she wants the result for a different nominal FDR value, he/she only needs to call the retest function *adafdr_retest* to generate the corresponding result. The retest function only contains the threshold rescaling step that takes almost no time. Such practice also maintains the monotonicity since the shape of the threshold is fixed and only the rescaling factor γ is changed for testing with different α s. We have added a notebook *demo_retest.ipynb* in the vignettes.

If the user chooses to run *adafdr_test* every time, however, the monotonicity may be slightly violated due to the inherent randomness of the algorithm. This, however, should not change the main result since the algorithm is stable as shown by the results above.

Regarding usability.

7. The authors have done a good job in making the AdaFDR software (github.com/martinjzhang/adafdr) and the analysis performed for the paper (github.com/martinjzhang/AdaFDRpaper) available on GitHub and pypi. However, I am

concerned with the limited usability of the method. Large scale statistical inference (i.e. hypothesis testing) in genomics and bioinformatics is often performed in R or with command line tools and not Python (note also that all other FDR controlling methods benchmarked in this paper are also implemented in R). The current software could be greatly improved if the authors provided an example for calling the AdaFDR Python function from R, e.g. using the "reticulate" package (<https://rstudio.github.io/reticulate/>) to access a wider audience of users.

Thank you for the comment. We have provided an R package using "reticulate" at <https://github.com/fxia22/RadaFDR>

Regarding interpretation.

8. Precise language needs to be used to describe the results. The descriptions and labels of the null and alternative hypothesis distribution plots (e.g. Figure 2C) are misleading. The plots show $f(x \mid \text{pval not significant})$ and $f(x \mid \text{pval significant})$, the conditional distributions of the covariate given that a test shows either weak (large p-value) or strong (small p-value) evidence for being significant. However, the plots are labeled as "null/alternative proportion", which suggests something else - namely, $p(h = 0 \mid x)$ and $p(h = 1 \mid x)$. This is made even more confusing as the discussion of these plots in the text make statements of $p(\text{pval significant} \mid x)$ and $p(\text{pval not significant} \mid x)$, e.g. "genes with higher expression levels are more likely to have significant associations" (page 3). First, and most importantly, "significant associations" should not be used interchangeably with tests being truly alternative (as in the plot label), This is not correct. Second, even if the two were the same, the conditioning is being flipped between the text/labels and the plots - further confusing the interpretation of these results. These details need to be clarified as it impacts the interpretation of the results presented in the paper. For example, while "genes with higher expression levels are more likely to have significant associations" (page 3), this **should not** be interpreted as genes having higher expression levels being more likely to be truly differential, as currently implied by Figure 2C and stated on page 4: "alternative hypotheses are more likely to occur when the expression levels are high." This is not a correct interpretation of the data. While more significant p-values occur when the expression levels are high, this can be due to reasons other than the association between alternative hypotheses and expression levels claimed in the text. Instead, it may simply be due to better power to detect differences when the expression levels are high.

Thank you for the comment. We have changed the label to "covariate distribution for null/alt", which indicates $P(x \mid \text{null})$ and $P(x \mid \text{alt})$. For the first concern regarding the equivalence between "significant hypothesis" and "true alternatives", we use hypotheses with p-values smaller than the BH threshold as an approximate of the alternative hypotheses. For the second concern regarding the flipping of the conditional probability, we agree with the reviewer. We have added the following clarification in the revision and have revised corresponding interpretations.

Covariate visualization via `AdaFDR_explore`

AdaFDR also provides a `FeatureExplore` function that can visualize the relationship between each covariate and the significance of hypotheses, **in terms of the marginal covariate distribution for the null hypothesis and the alternative hypothesis respectively**, as those shown in Figure 2c and Figures 3b-e. Let x_i be the univariate covariate under consideration and $h_i = 0/1$ indicate the ground truth (true null/alternative) for the i th hypothesis. Then here we are trying to estimate the conditional covariate distribution given the hypothesis label, i.e., $\mathbb{P}(x_i|h = 0)$ and $\mathbb{P}(x_i|h = 1)$. Noting that as a function of x_i ,

$$\frac{\mathbb{P}(x_i|h = 1)}{\mathbb{P}(x_i|h = 0)} \propto \frac{\mathbb{P}(x_i, h = 1)}{\mathbb{P}(x_i, h = 0)} = \frac{\mathbb{P}(h = 1|x_i)}{\mathbb{P}(h = 0|x_i)}.$$

The ratio between the two distributions can also be interpreted, up to a scale factor, as the the ratio of the hypothesis being true alternative/null given the covariate.

Minor Issues

- In general, the Introduction is awkwardly organized, with (somewhat redundant) subsections and a few mislabeled references. This should be cleaned up.

Thank you for the comment. We have cleaned up the Introduction accordingly.

- Throughout, "multiple hypothesis testing" is used to mean "multiple testing correction". This should be corrected.

Thank you for the comment. We have added a footnote on the first page as clarification:

¹Also known as multiple testing correction procedures.

- On page 2, mathematical notation is introduced without ever being mentioned again in the main text and can be dropped.

Thank you for the comment. We have dropped the notations P_i and \mathbf{x}_i on page 2 and only mention them in the Methods section.

- The descriptions of the various applications (e.g. RNA-seq differential expression analysis) should be made more precise. For example, "RNA-seq data" is used synonymously with differential gene expression analysis with RNA-seq data (page 2).

Thank you for the comment. We have revised the descriptions.

REVIEWERS' COMMENTS:

Reviewer #1 (Remarks to the Author):

The authors addressed my comments. There is no further comment.

Reviewer #2 (Remarks to the Author):

The authors have done an excellent job in clarifying the issues the reviewer had in the previous round. It is reassuring to see the replication results and additional analyses.

Reviewer #3 (Remarks to the Author):

I thank the authors for the additional work performed to address the concerns raised in my previous review. Most of my concerns have been addressed through the inclusion of BL in the comparisons, and additional analyses and clarifications to the text. I still have one point of concern.

The sensitivity of the AdaFDR method revealed in the response to point 5 ("Regarding stability") is an important point that needs to be made clear in the main manuscript. The results presented in the new Supplementary Figure S7 show that the results returned by the method can change not dramatically, but still noticeably (average replication rate of 92.4% across datasets). If I understand this correctly, if someone runs the method twice, the significant calls will only overlap ~90% between the runs. (Even if the authors set the random seed internally, this does not prevent a similar issue from arising when a user reorders the p-values or drops a single test from the set.) Currently, the stochasticity of the method and Supplementary Figure S7 do not appear to be referenced in the main paper. If it has not been mentioned, it should be stated (at the very least on page 6 in the Methods section) so that it is clear to the reader.

Aside from this point, I am happy with the changes made to the manuscript by the authors, and believe the method is a useful addition to the current literature of FDR-controlling methods.

I agree to have my name released.

Patrick Kimes, PhD
Postdoctoral Research Fellow
Department of Data Sciences, Dana-Farber Cancer Institute
Department of Biostatistics, Harvard TH Chan School of Public Health

Point-by-point response to referee's comments

We thank the reviewers for their thoughtful feedback.

Reviewer #1 (Remarks to the Author):

The authors addressed my comments. There is no further comment.

Thank you for the comment.

Reviewer #2 (Remarks to the Author):

The authors have done an excellent job in clarifying the issues the reviewer had in the previous round. It is reassuring to see the replication results and additional analyses.

Thank you for the comment.

Reviewer #3 (Remarks to the Author):

I thank the authors for the additional work performed to address the concerns raised in my previous review. Most of my concerns have been addressed through the inclusion of BL in the comparisons, and additional analyses and clarifications to the text. I still have one point of concern.

The sensitivity of the AdaFDR method revealed in the response to point 5 ("Regarding stability") is an important point that needs to be made clear in the main manuscript. The results presented in the new Supplementary Figure S7 show that the results returned by the method can change not dramatically, but still noticeably (average replication rate of 92.4% across datasets). If I understand this correctly, if someone runs the method twice, the significant calls will only overlap ~90% between the runs. (Even if the authors set the random seed internally, this does not prevent a similar issue from arising when a user reorders the p-values or drops a single test from the set.) Currently, the stochasticity of the method and Supplementary Figure S7 do not appear to be referenced in the main paper. If it has not been mentioned, it should be stated (at the very least on page 6 in the Methods section) so that it is clear to the reader.

Aside from this point, I am happy with the changes made to the manuscript by the authors, and believe the method is a useful addition to the current literature of FDR-controlling methods.

Thank you for the comment. We added a note regarding the algorithm stability in the Discussion section in the final paper:

step would produce an overly conservative result in this case. In addition, AdaFDR may produce slightly different results (< 10%) in two runs with different random seeds because of the random hypotheses splitting step (see Supplementary Figure 7 for more details). It is recommended to fix the random seed for better reproducibility. Nonetheless, in all cases the discoveries are valid in that the FDR is controlled.